# Altered Brain Metabolome Is Associated with Memory Impairment in the rTg4510 Mouse Model of Tauopathy

**DOI:** 10.3390/metabo10020069

**Published:** 2020-02-14

**Authors:** Mireia Tondo, Brandi Wasek, Joan Carles Escola-Gil, David de Gonzalo-Calvo, Clinton Harmon, Erland Arning, Teodoro Bottiglieri

**Affiliations:** 1Center of Metabolomics, Institute of Metabolic Disease, Baylor Scott & White Research Institute, Dallas, TX 75226, USA; MTondo@santpau.cat (M.T.); Brandi.WasekPatterson@BSWHealth.org (B.W.); clinton.r.harmon@gmail.com (C.H.); Erland.Arning@BSWHealth.org (E.A.); 2Servei de Bioquímica, Laboratori Sant Pau, Hospital de la Santa Creu i Sant Pau, 08041 Barcelona, Spain; 3Research Institute, Hospital de la Santa Creu i Sant Pau and CIBERDEM, Institute of Health Carlos III, 08041 Barcelona, Spain; jescola@santpau.cat; 4Institute of Biomedical Research of Barcelona (IIBB)—Spanish National Research Council (CSIC), Biomedical Research Institute Sant Pau (IIB Sant Pau) and CIBERCV, Institute of Health Carlos III, 08036 Barcelona, Spain; david.degonzalo@gmail.com

**Keywords:** metabolomics, Alzheimer’s disease, tauopathy, rTg4510 mice, cognitive impairment

## Abstract

Alzheimer’s disease (AD) is characterized, amongst other features, by the pathologic accumulation of abnormally phosphorylated tau filaments in neurons that lead to neurofibrillary tangles. However, the molecular mechanisms by which the abnormal processing of tau leads to neurodegeneration and cognitive impairment remain unknown. Metabolomic techniques can comprehensively assess disturbances in metabolic pathways that reflect changes downstream from genomic, transcriptomic and proteomic systems. In the present study, we undertook a targeted metabolomic approach to determine a total of 187 prenominated metabolites in brain cortex tissue from wild type and rTg4510 animals (a mice model of tauopathy), in order to establish the association of metabolic pathways with cognitive impairment. This targeted metabolomic approach revealed significant differences in metabolite concentrations of transgenic mice. Brain glutamine, serotonin and sphingomyelin C18:0 were found to be predictors of memory impairment. These findings provide informative data for future research on AD, since some of them agree with pathological alterations observed in diseased humans.

## 1. Introduction

The prevalence of dementia doubles every 5 years over the age of 65 [1]. Alzheimer’s disease (AD) is a progressive and fatal neurodegenerative disorder and the most common form of dementia, accounting for 60–80% of all dementia cases [1]. AD is clinically characterized by progressive memory loss, mood changes, problems with communication and reasoning, and eventual loss of independent living. AD is characterized by the pathologic accumulation of extracellular amyloid beta (Aβ) and abnormally phosphorylated tau filaments in neurons that lead to senile plaques and neurofibrillary tangles (NFT), respectively, following a specific spatial and temporal pattern [2,3,4]. All disorders that cause the accumulation of tau protein are called tauopathies and include AD and frontotemporal dementia with Parkinsonism (FTDP), amongst other neurodegenerative diseases. The abnormally hyperphosphorylated tau is considered one of the main hallmarks of AD [2]. However, the molecular mechanisms by which the abnormal processing of tau leads to neurodegeneration and cognitive impairment remain unknown.

Various mouse models that recapitulate the neuropathological and phenotypic features of AD are widely used in experimental studies to investigate the pathophysiological role of Aβ plaques and NFT in AD. These murine models contain either amyloid-based transgenes that overexpress the amyloid precursor protein (APP) protein, tau protein and/or its processing enzymes (presenilins or mutant microtubule-associated protein tau (MAPT) expressers) or a combination of both. Advanced metabolomic analysis has been applied to transgenic AD models. Although several reports have revealed significant differences in the levels of metabolites in murine models that overexpress the amyloid protein [5,6,7,8,9,10], no such studies have been performed in a tauopathy mouse model.

The most frequent mutation in the tau gene located on chromosome 17q21 in humans is P301L mutant tau linked to FTDP-17 specifically in cortex, limbic system and basal ganglia. The high levels of transgenic tauP301L expression in rTg4510 mice induce the age-dependent development of the three major pathological hallmarks of human tauopathy: memory impairment, neurofibrillary tangles and neuron loss similar to that observed in AD. The model is characterized by spatial memory deficits, the formation of a distinct 64 kDa abnormally hyperphosphorylated isoform of tau and an increase in NFT, with a rapidly progressive neuronal loss in the hippocampus by 5.5 months of age [11,12,13].

Metabolomic techniques can comprehensively assess disturbances in metabolic pathways that reflect changes downstream from genomic, transcriptomic and proteomic systems [14,15]. This technique and the information obtained have considerable potential as a discovery platform for identifying novel diagnostic biomarkers and therapeutic targets for AD and other neurodegenerative diseases. In addition, biomarkers may be used to predict the risk of dementia in the preclinical stage, as recently discussed in a systematic review on metabolomics in the development and progression of dementia [16]. Considerable evidence suggests that several metabolites, including lipids, amino acids and steroids, are associated with cognitive decline.

In the present study, we undertook a targeted metabolomics approach to determine a total of 187 prenominated metabolites in brain cortex tissue from wild type and rTg4510 animals, in order to establish the association of metabolic pathways with cognitive impairment.

## 2. Materials and Methods

### 2.1. Animals

rTg4510 mice were bred by crossing mice expressing the responder mutant P301L transgene with mice expressing the tetracycline-dependent transcription activator (tTa) transgene, as previously described [12]. F1 littermates were used in all experiments. Briefly, the responder transgene consists of a tetracycline operon-responsive element placed upstream of a cDNA encoding human tau with four microtubule binding repeats (4R tau) and the P301L mutation. The activator transgene contains the tet-off open reading frame placed downstream of Ca^2+^/calmodulin kinase II promoter elements. Mice were genotyped by the analysis of tail DNA using tau cDNA-specific primers to exon 1 and exon 5, and primers specific to tetracycline trans-activator. Double negative mice were used as wild type (WT) controls. Twenty WT (10 males and 10 females) and 16 rTg4510 transgenic mice (7 males and 9 females) were used for behavioral tests and metabolomic analyses. All experimental mice were bred in a pathogen-free environment, maintained in a temperature-controlled animal facility on a 12-h light/dark cycle and allowed access to food and water ad libitum. At 4 months of age, behavioral studies were performed, and the mice were sacrificed by microwave fixation. The use of microwave fixation prevents postmortem metabolism by the inactivation of enzyme activity in brain tissue. Briefly, mice (25–35 g) were placed in a cylindrical Perspex holder, which prevents any movement and keeps the head of the mouse in a fixed position. The holder and mouse were placed in the chamber of the microwave system (Model TMW-4012c, Muromachi Kikai Co. Ltd., Tokyo, Japan) and exposed to the microwave beam at an intensity of 6 kW for 0.9 s. Afterwards, brain tissue was removed for regional dissection. This technique raises the brain tissue temperature to 85–95 °C, immediately stopping any enzyme and metabolic activity, while preserving the structure and integrity of the brain. Harvested tissues were stored at −80 °C until the metabolic and protein analyses. All experiments with mice were performed in accordance with protocols approved by the Institutional Animal Care and the Use Committee at Baylor Scott & White Research Institute (protocol code A17-002, approval date 21 July 2017).

### 2.2. Behavioral Tests

The Morris Water Maze (MWM) test was used to evaluate the spatiotemporal function. All experiments were carried out in a pool (173 cm in diameter and 76 cm deep) that was filled with room temperature water (22–24 °C). Nontoxic white paint was added until the water was opaque, to contrast with the mouse. The water maze pool had a visible platform, and in the margins 3 fixed points were defined from which the animal was released to find the platform. In our experimental design, trials were performed in triplicate, with a 2-min interval between them, on each day of testing. Mice were allowed to remain on the platform for 15 s before being removed. The training process was repeated for 4 consecutive days with the platform visible. The experiment was repeated the following 5 days with the platform submerged 0.5 cm under water. This time, the mouse had to locate the platform by navigation from spatial memory clues (latency time). On the last day (probe trial), the platform was totally removed, and the animals were released to the pool from one point. The path in the target quadrant was monitored for 30 s and associated with each of the studied metabolites. All sessions were tracked using an overhead camera (HVS tracking system, Hampton, FL, USA) interfaced to ActiMetrics software version 2.6 (Colbourn Instruments, Allentown, PA, USA).

### 2.3. Preparation of Brain Extracts

For the metabolites analyses, left-brain cortex samples were prepared for analysis by sonication for 30 s (3 cycles of 10 s) with 4 volumes (wt/vol) of 85% ethanol and 15% phosphate buffered saline. Following centrifugation at 14,000 rpm for 5 min at 4 °C, clear extracts were stored overnight at −80 °C, and a metabolomic analysis was performed the following day.

For the protein analyses, right-brain cortex samples were homogenized by sonication for 30 s (3 cycles of 10 s) in 10 volumes (wt/vol) of hot 1% SDS solution containing protease inhibitor (cOmplete Mini, Sigma Aldrich) and phosphatase inhibitor (PhosSTOP, Sigma Aldrich), followed by incubation for 10 min at 90 °C. Aliquots were then incubated on ice for 10 min and centrifuged at 14,000 rpm for 5 min at 4 °C. The protein content was quantified by a BCA protein assay, and the samples were analyzed immediately or kept frozen at −80 °C for future analyses.

### 2.4. Metabolomics Study

For the targeted metabolomics study, quantitative and semiquantitative mass spectrometry based metabolomic profiling was performed using the Biocrates Absolute*IDQ* p180 (Biocrates, Life Science AG, Innsbruck, Austria), as previously described [17,18]. The Absolute*IDQ* p180 kit provided the simultaneous quantification of 21 amino acids, 21 biogenic amines, the sum of hexoses (including glucose), 40 acylcarnitines, 15 sphingolipids (SPHs), 14 lysophosphatidylcholines (lyso-PC) and 76 phosphatidylcholines (PC). The samples were processed according to the manufacturer’s instructions and analyzed on a triple-quadrupole 5500 QTRAP mass spectrometer (Sciex, Foster City, CA, USA) coupled with a Prominence Nexera ultrahigh pressure liquid chromatography system (Shimadzu, Kyoto, Japan). Briefly, 10 µL of postmortem brain extract was loaded in a 96-well format, which included calibration standards. This assay is based on phenylisothiocynate derivatization in the presence of isotopically labeled internal standards. Amino acids and biogenic amines were analyzed by liquid chromatography (LC) tandem mass spectrometry and the other metabolites by flow injection tandem mass spectrometry. The identification and quantification were performed based on internal standards and multiple reaction monitoring (MRM) detection. Data was collected using the Analyst software version 1.6. (Sciex, Foster City, CA, USA) and uploaded and processed in the Met*IDQ* software (Biocrates Life Sciences, Vienna, Austria). The final metabolite concentrations were calculated and expressed as µmol/mg of tissue.

### 2.5. Determination of Protein Expression

Total tau and phosphorylated tau (p-Tau) were measured by the Simple Western System™ using the Wes instrument (ProteinSimple^®^, San Jose, CA, USA). The Wes system is based on the separation of proteins by size, using capillary electrophoresis with immuno-antibody detection, and provides a greater level of sensitivity and accuracy compared to the traditional gel western blot methods. Samples, blocking reagents, primary antibody, secondary antibody and fluorescence substrate were loaded onto a 24-well plate according to the Wes ProteinSimple^®^ user manual for automated analysis. The following primary antibodies were used for the detection of brain proteins: beta-actin (#ab6276, AbCcam), total tau (Tau-5, #1261887A, Invitrogen), tau phosphorylated at Ser202 (CP13) and tau phosphorylated at Ser396/Ser404 (PHF-1) were gifted from Dr. Peter Davies (Albert Einstein College of Medicine, New York, NY, USA). Compass software (ProteinSimple^®^, San Jose, CA, USA) was used to process data and to calculate the area under the curve of eluted fluorescence-labeled proteins.

### 2.6. Statistical Analysis

Data normality was evaluated using the Kolmogorov–Smirnov test. Non-normally distributed variables were logarithmically transformed to account for nonlinearity. Continuous variables were compared between study groups using a Student’s t-test for independent samples and a two-way ANOVA, followed by a Bonferroni post hoc test. Correlations between variables were analyzed using Pearson’s correlation analysis. The results were presented using Pearson’s correlation coefficient (ρ). Backward stepwise linear regression analyses were performed to identify the best model associated with path in target. Path in target was entered as a dependent variable, and the best candidate of each subgroup of metabolites was subsequently entered as an independent variable. Data were expressed as standardized beta (β) and coefficient of determination (R^2^). Data were processed in Metaboanalyst 4.0 to generate heat map visualizations, the false discovery rate (FDR) and the principal component analysis (PCA) [19], as well as to determine the metabolites that can differentiate between WT and transgenic mice. Statistical analyses were performed with the GraphPad Prism software (GraphPad Software Inc., La Jolla, CA, USA) and the statistical software R (www.r-project.org).

## 3. Results

### 3.1. Memory Impairment in the Transgenic Mice Model rTg4510 of Tauopathy

Latency time was first evaluated in male and female WT and rTg4510 mice. MWM data did not reveal differences between males and females within each genotype (data not shown), and therefore mice were no longer grouped according to sex. When WT and rTg4510 mice where compared, all animals improved their spatial memory of the location of the escape platform in the reference memory version of MWM during training. However, the two groups differed significantly in their latency time for days 1, 2 and 3 (*p* < 0.05) (Figure 1A). During the cued test, the groups differed strongly when searching for the escape platform for days 6, 7, 8 and 9 (*p* < 0.0001), as the times of rTg4510 mice were significantly longer than those of the non-Tg littermates (Figure 1A). In line with these findings, rTg4510 mice spent significantly less time in the target quadrant (*p* < 0.0001) (Figure 1B). The average speed did not differ between rTg4150 mice and WT mice (data not shown), indicating no impairment between the two groups of mice in the ability to swim during the test. As expected, the analysis of total tau and phosphorylated tau epitopes (CP13 and PHF-1) in rTg4510 mice were significantly higher compared to WT mice (Figure 2A–C).

### 3.2. Altered Brain Metabolome Profile in rTg4510 Mice

In the P180 platform (Biocrates, Life Science, Vienna, Austria), we analyzed 187 metabolites in brain samples from WT and rTg4510 mice. A heat map was used to visualize the metabolite signature from both genotypes (Figure 3). Hierarchical clustering suggested different metabolite profiles in WT and rTg4510 mice.

We then evaluated how brain metabolites were significantly changed in the cortex of rTg4510 mice. Within each class of compounds there were significant changes between WT and rTg4510 mice (Table 1).

Twenty-one PCs (27.6%) were decreased in transgenic mice when compared to WT (PC aa C30:0, PC aa C32:0, PC ae C32:1, PC aa C42:4, PC aa C32:3, PC aa C34:1, PC aa C36:6, PC ae C32:2, PC aa C36:0, PC ae C30:0, PC ae C34:0, PC ae C34:3, PC ae C36:0, PC aa C36:1, PC ae C34:1, PC aa C42:5, PC aa C36:2, PC aa C38:0, PC ae C30:1, PC ae C36:1, PC aa C28:1). Nine amino acids (42.9%) showed changes and seven were increased (arginine, glutamine, leucine, proline, tryptophan, valine and histidine), whereas asparagine and taurine were significantly decreased in transgenic mice when compared to WT. Furthermore, two biogenic amines (9.5%) showed significant increase in transgenic mice when compared to WT (putrescine and serotonin). Seven SPHs (46.7%) were decreased in transgenic mice when compared to WT (SM OH C16:1, SM OH C22:1, SM OH C22:2, SM OH C24:1, SM C26:1, SM C18:0, SM C20:2). Regarding acylcarnitines, two (5%) were significantly increased (C3-OH and C4:1) in transgenic mice when compared to WT. We also assessed the main effect of genotype and sex and the potential interaction between them. The results demonstrated that most of the metabolomic changes in WT mice were independent of sex except for creatinine and t4-OH-Pro, and lysoPC a C26:0 and t4-OH-Pro for the transgenic mice (data not shown).

### 3.3. Brain Metabolic Profile and Memory Impairment Associations

Cognition-related metabolites that significantly correlated with percentage of path in target are shown in Table 2. For the PC group, two of them were negatively correlated: lysoPC a C16:0 (ρ = −0.423) and lysoPC a C17:0 (ρ = −0.422), whereas ten positively correlated with percentage of path in target: PC aa C30:0 (ρ = 0.477), PC aa C32:0 (ρ = 0.451), PC aa C32:2 (ρ = 0.348), PC aa C32:3 (ρ = 0.358), PC aa C34:1 (ρ = 0.346), PC aa C36:6 (ρ = 0.358), PC aa C42:4 (ρ = 0.447), PC ae C32:1 (ρ = 0.449), PC ae C34:0 (ρ = 0.499), PC ae C36:0 (ρ = 0.468). In the amino acid group, glutamine and ornithine were negatively correlated (ρ = −0.443 and ρ = −0.354, respectively), whereas alanine, asparagine and methionine were positively correlated with percentage of path in target (ρ = 0.404, ρ = 0.387 and ρ = 0.35, respectively). For the biogenic amines group, three of them were negatively correlated with percentage of path in target: Ac-Orn (ρ = −0.338), putrescine (ρ = −0.49) and serotonin (ρ = −0.54). For the SPHs group, two of them correlated positively with percentage of path in target: SM (OH) C16:1 (ρ = 0.415) and SM C18:0 (ρ = 0.44). Finally, for the acylcarnitines group, three of them correlated negatively: C10:1 (ρ = −0.376), C16:2 (ρ = −0.378) and C5-OH(C3-DC-M) (ρ = −0.363), and two of them correlated positively with percentage of path in target: C10:2 (ρ = 0.348) and C5:1-DC (ρ = 0.336).

### 3.4. Brain Glutamine, Serotonin and SM C18:0 Are Predictors of Memory Impairment in rTg4510 Mice

The metabolites that showed the highest correlation with percentage of path in target quadrant within each category were subjected to a multivariate statistical analysis in order to determine the main metabolic abnormalities associated with cognitive impairment by means of percentage of path in target quadrant. A linear regression analysis revealed a negative association between glutamine (β = −0.403, *p* = 0.004) and serotonin levels (β = −0.325, *p* = 0.021) and percentage of path in target quadrant, and a positive association between SM C18:0 levels (β = 0.380, *p* = 0.007) and percentage of path in target quadrant (Table 3). The multivariate model, including glutamine, serotonin and SM C18:0, explains 52% of the physiological effect (R^2^ =0.521).

Based on these findings, PCA was employed to visually discriminate between rTg4510 (circles) and WT (squares) mice considering glutamine, serotonin and SM C18:0. The score plots demonstrate that it was possible to visibly discern WT from transgenic mice in the brain even though some degree of overlap existed (Figure 4).

## 4. Discussion

Metabolomic analysis offers a high potential for the discovery of novel biomarkers for diagnosing and elucidating the underlying mechanisms of neurodegenerative diseases. In this study, using a targeted metabolomic platform, we identified specific metabolite changes consequential to the development of AD-like pathology in brain cortex tissue from 4 months-old rTg4510 mice. This model showed the expected increase in taupathology and decline in cognitive function. To our knowledge, this is the first targeted metabolomic analysis to investigate metabolic alterations in brain tissue from rTg4510 mice associated with cognitive deficiency. Similar studies have been performed in other models of AD focusing on Aβ pathology, such as APP/PS1 mice [5,8,20,21] or TASTPM mice [7]. However, the correlation between metabolomic alterations and cognitive impairment has not been addressed. The MWM test is a classical behavioral tool used to investigate learning and spatial memory in various mouse models of AD. In our study, we used a targeted approach to determine 187 metabolites including amino acids, biogenic amines, phospholipids and acylcarnitines. It is known that rTg4510 mice start developing significant cognitive impairment at 4 months of age [11,12]. Therefore, a metabolomic approach at this time period was chosen to identify possible metabolic markers at an early stage in the course of AD. The study design utilized a targeted comprehensive analysis of metabolites in conjunction with microwave fixation and associating metabolic alterations with a physiological marker of cognitive impairment, i.e., the measure of percentage of path in target on the last day of the MWM test. It should be noted that male and female rTg4510 mice showed similar pathology and learning acquisition and/or spatial memory. In line with these findings, several previous studies reported no sex differences in rTg4510mice [11,12,22,23,24]. However, one study showed a more aggressive memory impairment in 5.5 months-old rTg4510 female mice when compared to their male littermates [25]. In our study, the earlier age of intervention could attenuate sex-dependent differences.

An original and novel aspect of this study is the use of microwave, using the Muromachi Microwave Fixation System to sacrifice mice. In comparison to other traditional methods of sacrificing rodents, such as CO_2_ asphyxiation or cervical dislocation, microwave fixation offers a real time approach to the whole brain metabolome and proteome, as previously reported by our group [26]. This procedure allows for the rapid cessation of metabolic activity (< 0.9 s), avoiding postmortem metabolic changes and thereby enabling an accurate determination of brain tissue metabolites.

The targeted metabolomic approach was able to reveal significant differences in metabolite concentrations of transgenic mice. Hence, we found a higher number of changes in the SPHs (7 of 15, 46.7%), followed by amino acids (9 of 21, 42.9%), PCs (21 of 76, 27.6%), biogenic amines (2 of 21, 9.5%) and finally acylcarnitines (2 of 40, 5%). These changes in the metabolomic profile were sex-independent for all metabolites except for four, confirming that metabolomic differences between WT and transgenic mice were largely independent of sex at that age period. Overall, changes in sphingolipids and amino acids were the most significant between rTg4510 and WT mice. Almost half of all SPHs measured in rTg4510 mice were significantly decreased at the age of 4 months. In AD, perturbed sphingomyelin (SM) metabolism has been proposed as a pivotal event in the dysfunction and degeneration of neurons, with increased SM levels in the cerebrospinal fluid (CSF) [27] and decreased in blood or plasma [28,29,30]. However, postmortem human brain analyses have rendered divergent results, showing increased [31] or decreased [32] total concentrations of SM. Previous studies with APP/PS1 mouse brain found significantly elevated levels of more than one third of all SPHs [8]. SPHs are precursors for ceramide production, mainly by the action of SM degrading enzyme neutral sphingomyelinase. Ceramide accumulation induces apoptosis and seems to worsen neurodegeneration by increasing Aβ biosynthesis [33]. Therefore, neutral sphingomyelinase could be promoting SPHs degradation and ceramide accumulation, and hence promoting apoptotic cell death. Altogether, these findings suggest a major role of SPHs in the dysfunctional management of cellular membranes in transgenic tauopathy mice, which requires further investigation. On the other hand, more than one third of amino acids measured in rTg4510 mice were significantly elevated at the age of 4 months. Previous studies in other genetically-engineered mice models of AD, such as APP/PS1, CRND8 and TASTPM mice, reported increased concentrations of several amino acids in the brain [5,7,9,10]. These observations may indicate an important deregulation of the transport of amino acids across the blood brain barrier, thereby indicating that disturbances in amino acid homeostasis might play a critical role in the pathogenesis of AD in rTg4510 mice.

When studying metabolic profile and memory impairment associations, several metabolites (27 out of 187) significantly correlated (either positively or negatively) with percentage of path in target as a biological marker. The metabolites that showed the highest correlation with percentage of path in target quadrant were subjected to multivariate statistical analysis in order to identify metabolites that could be potential markers of the disease. The results showed that glutamine and serotonin presented a negative association with percentage of path in target, whereas SM C18:0 presented a positive one, leading to the conclusion that these three metabolites may be potential independent predictors of memory impairment in rTg4510 mice.

It is well known that neurotransmission plays a crucial role in the pathogenesis of neuropsychiatric behaviors in AD [34]. Serotonin is an important neurotransmitter that participates in the modulation of memory formation, mood and emotional states [35]. It is unclear whether the atrophy of serotonergic neurons obtained in the brains of AD patients is a consequence of the general neurodegeneration or whether it contributes to disease progression. We observed increased serotonin brain concentrations in tauopathy mice, which contrast with concentrations previously reported in plasma or brain tissue from AD subjects [36,37]. Nonetheless, another study performed with APP/PS1 mice reported an unexplained 70% increase in plasma serotonin [8]. The accumulation of serotonin in experimental mice models of AD deserves further investigation.

With respect to glutamine, astrocytes convert glutamate into glutamine by glutamate synthetase [38]. Astrocytes provide the source of glutamine that is taken up and utilized by neurons for conversion into glutamate by glutaminase, which is then converted to the main inhibitory neurotransmitter, gamma-amino butyric acid. The loss of astrocytic processes and the inability to traffic glutamate transporters leads to glutamate imbalance [39,40,41]. Previous studies of glutamate and glutamine levels in CSF have shown controversial findings. Some authors reported higher CSF glutamate values in AD patients than in controls [42,43], whereas others reported decreased values [44,45] or no changes at all [46,47,48]. A recent study using C13-glutamine performed in APP/PS1 mice suggests that reduced glutamine uptake and impaired oxidative glutamine metabolism could be very early markers of AD pathogenesis, as they precede the amyloid plaque formation in this model [49]. This hypothesis may be plausible in the rTg4510 mouse of the AD model at an early pathological stage. Furthermore, a recent study reported increased glutamate and glutamine levels in the CSF of patients with AD, suggesting that increased glutamate may cause the build-up of Aβ peptides in the AD brain, underlying, at least in part, cognitive and mild alterations in AD at early stages of dementia [50].

Although no studies have investigated the association between SM C18:0 and AD, the association of sphingomyelins and ceramide metabolism could affect the progression of cognitive impairment in rTg4510 mice. It should be noted that the deposition of senile plaques containing Aβ peptides and the formation of neurofibrillary tangles are mainly localized in medial temporal lobe structures, specifically the cortex and the hippocampus [51]. While numerous studies in transgenic models of AD have been performed to study the whole brain, other studies have focused on individual brain areas [10,52,53,54,55,56], as metabolic perturbations induced by AD-type disorders can be region-specific in the brain [57,58]. Most of the findings from these studies suggest that the cortex and the hippocampus are the most sensitive regions during early-stage AD [53,59]. Moreover, it was determined that tangle formation pathology is first observed in the cortex area and progresses into the hippocampus and limbic structures with the increase of age [11].

In conclusion, mutant rTg4510 transgenic mice can be differentiated from wild type littermates by metabolomic analysis of cortical brain samples at an early age. These findings provide informative data for future metabolomic studies of AD and other human neurodegenerative disorders where the accumulation of tau protein is a neuropathological feature.

## Figures and Tables

**Figure 1 metabolites-10-00069-f001:**
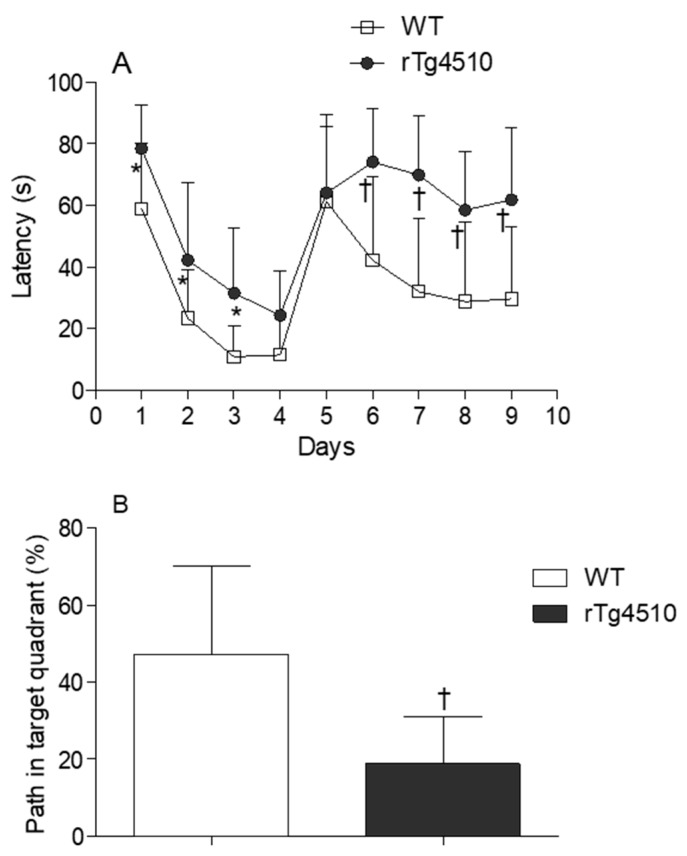
(**A**) Water maze test for the evaluation of spatial memory in 4 months-old wild type (WT) (*n* = 20) and rTg4510 (*n* = 16) mice. (**B**) Percentage of path in the target quadrant in 4 months-old WT (*n* = 20) and rTg4510 (*n* = 16) mice.

**Figure 2 metabolites-10-00069-f002:**
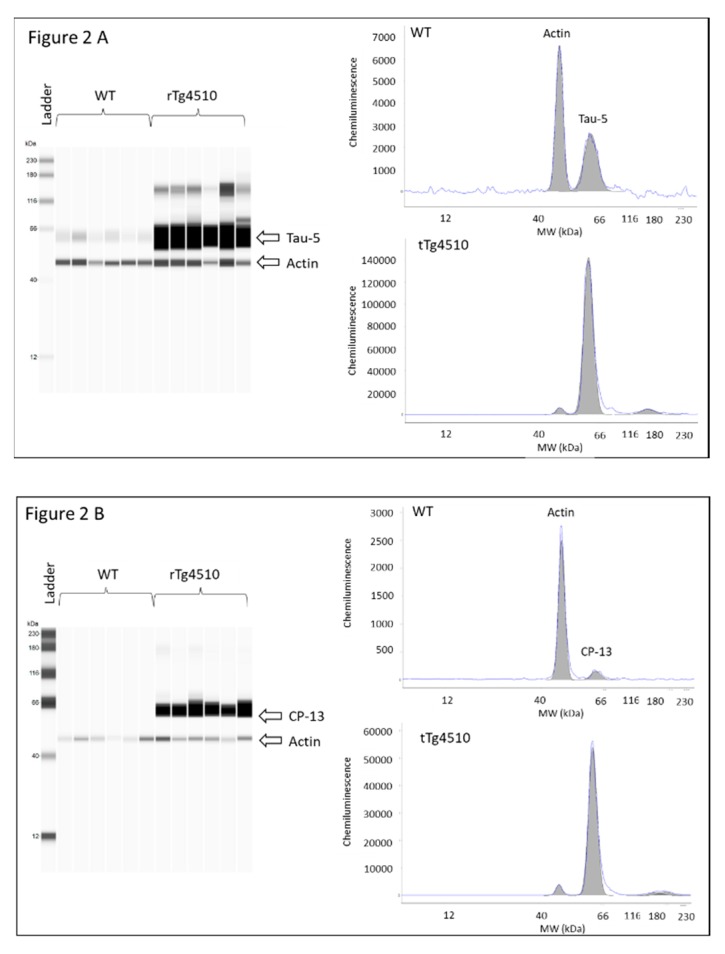
Levels of total tau and phosphorylated tau in brain cortex tissue from 4 months-old WT and rTg4510 mice; (**A**) total tau; (**B**) CP13; and (**C**) PHF1. Left panel virtual western blot; right panel representative pherogram of brain cortex from a WT and rTg4510 mouse. Similar profiles were found for all WT and rTg4510 mice used in the study.

**Figure 3 metabolites-10-00069-f003:**
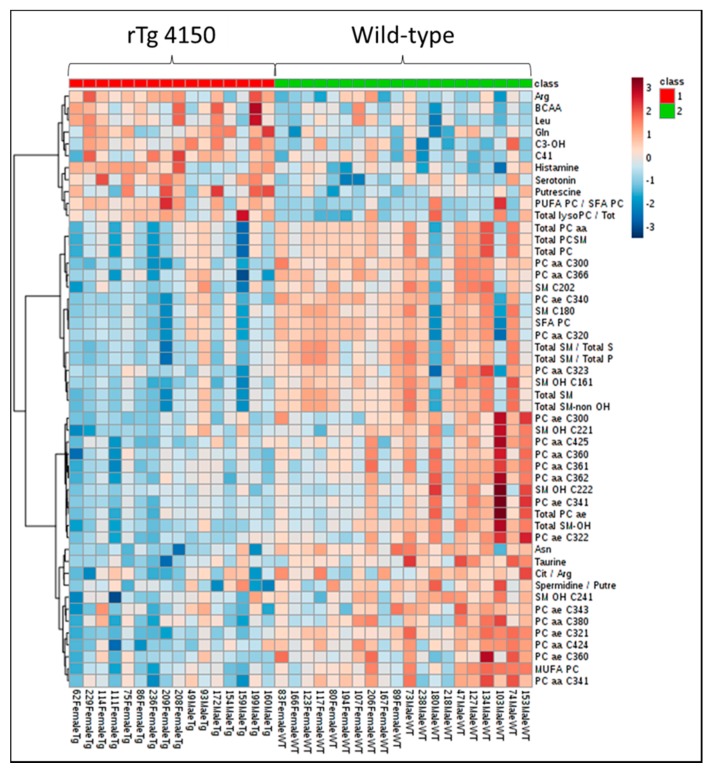
Profile of metabolites in brain samples from WT and rTg4510 mice. The heat map illustrates the levels of the top 50 metabolites that are significantly altered. The color scale on the right side of the map indicates increasing concentrations (red spectra) or decreased concentrations (blue spectra).

**Figure 4 metabolites-10-00069-f004:**
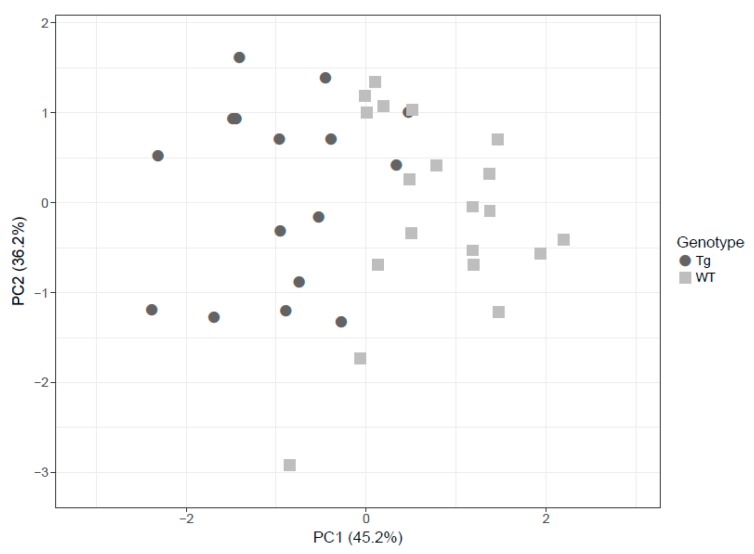
The principal component analysis (PCA) decomposition of the three selected metabolites (glutamine, serotonin and SM C18:0) could distinguish WT mice (circles) and rTg4510 mice (triangles).

**Table 1 metabolites-10-00069-t001:** Significant changes in cortical metabolites of non-Tg (WT) and rTg4510 mice. Cortices from four months-old WT (*n* = 20) and rTg4510 (*n* = 16) mice were evaluated. Data are presented as mean ± SD of µmol/mg tissue; *p*-value represents significance.

Metabolite	WT	rTg4510	t.stat	*p*-Value	FDR
Glycerophospholipids					
PC aa C30:0	3.71 ± 0.54	2.47 ± 0.45	−7.5081	<0.0001	0.0000012
PC aa C32:0	101.9 ± 22.4	75.5 ± 15.0	−3.2926	0.0023	0.0144960
PC ae C32:1	1.66 ± 0.28	1.11 ± 0.147	−8.0568	<0.0001	0.0000005
PC aa C42:4	0.146 ± 0.02	0.113 ± 0.018	−5.0941	<0.0001	0.0002996
PC aa C32:3	0.228 ± 0.050	0.178 ± 0.022	−3.3024	0.0023	0.0144960
PC aa C34:1	146.9 ± 9.2	136.9 ± 7.1	−3.577	0.0011	0.0082230
PC aa C36:6	0.295 ± 0.037	0.239 ± 0.045	−4.0477	0.0003	0.0032613
PC ae C32:2	0.183 ± 0.037	0.143 ± 0.015	−4.4828	0.0001	0.0014143
PC aa C36:0	1.66 ± 0.94	0.815 ± 0.266	−4.3839	0.0001	0.0017034
PC ae C30:0	0.144 ± 0.024	0.118 ± 0.011	−4.2498	0.0002	0.0021383
PC ae C34:0	1.23 ± 0.35	0.687 ± 0.244	−4.9941	<0.0001	0.0003672
PC ae C34:3	0.154 ± 0.025	0.13 ± 0.026	−2.9554	0.0056	0.0264610
PC ae C36:0	0.358 ± 0.075	0.296 ± 0.036	−3.1638	0.0033	0.0171870
PC aa C36:1	62.9 ± 10.3	51.1 ± 5.4	−4.2241	0.0002	0.0021763
PC ae C34:1	9.87 ± 2.93	7.33 ± 0.62	−4.0578	0.0003	0.0032613
PC aa C42:5	0.180 ± 0.039	0.141 ± 0.020	−3.8824	0.0005	0.0044484
PC aa C36:2	41.7 ± 7.9	34.1 ± 3.8	−3.8131	0.0006	0.0050945
PC aa C38:0	0.340 ± 0.044	0.301 ± 0.036	−2.9493	0.0057	0.0264610
PC ae C30:1	0.047 ± 0.012	0.038 ± 0.008	−2.9132	0.0063	0.0284540
PC ae C36:1	3.54 ± 1.56	2.52 ± 0.46	−2.8434	0.0075	0.0333190
PC aa C28:1	0.093 ± 0.026	0.0763 ± 0.007	−2.7483	0.0095	0.0399800
**Amino acids**					
Arg	11.1 ± 3.1	15.8 ± 4.3	3.8875	0.0004	0.0044484
Asn	24.7 ± 2.8	19.4 ± 2.5	−5.9583	0.00001	0.0000563
Gln	1739 ± 295	2162 ± 435	3.379	0.0018	0.0123810
Tau	579 ± 29	541 ± 22	−4.3707	0.0001	0.0017034
Leu	14.3 ± 4.0	20.0 ± 7.2	3.2244	0.0028	0.0153300
Pro	23.4 ± 3.8	27.4 ± 4.8	2.7764	0.0089	0.0379620
Trp	5.56 ± 1.21	6.90 ± 1.77	2.6795	0.0113	0.0457270
Val	26.3 ± 4.4	31.4 ± 6.8	2.7782	0.0088	0.0379620
Hist	2.18 ± 0.57	3.0 ± 0.78	3.4083	0.0017	0.0118830
**Biogenic Amines**					
Putrescine	1.26 ± 0.20	1.65 ± 0.51	3.0893	0.0040	0.0191700
Serotonin	1.33 ± 0.26	1.66 ± 0.32	3.3716	0.0019	0.0123810
**Sphingolipids**					
SM OH C16:1	0.220 ± 0.038	0.154 ± 0.024	−6.3622	<0.0001	0.0000225
SM OH C22:1	0.090 ± 0.038	0.059 ± 0.020	−3.5462	0.0012	0.0086668
SM OH C22:2	0.122 ± 0.091	0.066 ± 0.012	−3.1426	0.0035	0.0174490
SM OH C24:1	0.134 ± 0.020	0.112 ± 0.021	−3.1212	0.0037	0.0180030
SM C26:1	0.05 ± 0.03	0.032 ± 0.015	−2.7332	0.0099	0.0407570
SM C18:0	30.4 ± 8.2	20.8 ± 5.2	−3.6524	0.0009	0.0074096
SM C20:2	0.123 ± 0.029	0.091 ± 0.024	−3.603	0.0010	0.0079330
**Acylcarnitines**					
C3-OH	0.181 ± 0.02	0.2 ± 0.016	3.2767	0.0024	0.0147300
C4:1	0.016 ± 0.002	0.019 ± 0.003	3.1925	0.0030	0.0162960

**Table 2 metabolites-10-00069-t002:** Correlation between metabolites and the percentage of path in target. The animals used for these analyses were 4 months-old WT (*n* = 20) and rTg4510 (*n* = 16) mice. ρ represents Pearson’s correlation coefficient; *p*-value represents significance.

Metabolite		lysoPC a C16:0	lysoPC a C17:0	PC aa C30:0	PC aa C32:0	PC aa C32:2	PC aa C32:3	PC aa C34:1	PC aa C36:6	PC aa C42:4	PC ae C32:1	PC ae C34:0	PC ae C36:0
**% Path in target**	**ρ**	−0.423	−0.422	0.477	0.451	0.348	0.358	0.346	0.358	0.447	0.449	0.499	0.468
	***p*** **-value**	0.010	0.010	0.003	0.006	0.038	0.032	0.039	0.032	0.006	0.006	0.002	0.004
**Amino acids**		**Ala**	**Asn**	**Gln**	**Met**	**Orn**							
**% Path in target**	**ρ**	0.404	0.387	−0.443	0.35	−0.354							
	***p*** **-value**	0.014	0.020	0.007	0.037	0.034							
**Biogenic amines**		**Ac-Orn**	**Putrescine**	**Serotonin**									
**% Path in target**	**ρ**	−0.338	−0.49	−0.54									
	***p*** **-value**	0.044	0.002	0.001									
**Sphyngolipids**		**SM (OH) C16:1**	**SM C18:0**										
**% Path in target**	**ρ**	0.415	0.44										
	***p*** **-value**	0.012	0.007										
**Acylcarnitines**		**C10:1**	**C10:2**	**C16:2**	**C5-OH (C3-DC-M)**	**C5:1-DC**							
**% Path in target**	**ρ**	−0.376	0.348	−0.378	−0.363	0.336							
	***p*** **-value**	0.024	0.037	0.023	0.030	0.045							

**Table 3 metabolites-10-00069-t003:** Multivariate linear regression model. The animals used for these analyses were 4 months-old WT (*n* = 20) and rTg4510 (*n* = 16) mice; *p*-value represents significance; β represents standardized beta; R^2^ represents coefficient.

Model	Metabolite	β	*p*-Value	R^2^
	**Gln**	−0.403	0.004	0.521
	**Serotonin**	−0.325	0.021	
	**SM C18:0**	0.380	0.007

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
