# Peer review of "Altered Brain Metabolome Is Associated with Memory Impairment in the rTg4510 Mouse Model of Tauopathy"

_metabolites, 2020, doi:10.3390/metabo10020069_

Round 1
Reviewer 1 Report
The experiment is very well planned and conducted in accordance to highest standards in the field.
I have no objections, just several questions and suggestions.
1) What do you think you would get if you analyzed other parts of the brain? Was the decision to analyze cortices mainly based on the data that tangles develop in cortex first you cite in the discussion?
2) Did you also do exploratory PCA with all the metabolites? If you only did PCA with the 3 selected components maybe exploratory analysis of all the metabolites and complementary step-wise dimension reduction would reveal the biochemical pathways involved with the effect you observed.
3) Do you have any other behavioral data apart from what is presented in Fig 1. (velocity, distance, etc.)? Behavioral protocols differ greatly in the literature so I suggest you describe your behavioral protocol a little bit more in detail. Were the mice removed once they found the platform? What was the cumulative time spent in the pool for all the groups? What do you think why there was increased latency on the first day of the protocol and how are the latency timed distributed in the first trial?
Author Response
1) What do you think you would get if you analyzed other parts of the brain? Was the decision to analyze cortices mainly based on the data that tangles develop in cortex first you cite in the discussion?
Response: It is likely that there are changes in other brain regions where Tau and phosphorylated forms of Tau accumulate, such as the hippocampus. However, we had insufficient amount of tissue from this small regions that was required and used for other analysis. Future studies should explore metabolomic analysis in other brains regions. We have included a sentence that refers to this on line 411.
2) Did you also do exploratory PCA with all the metabolites? If you only did PCA with the 3 selected components maybe exploratory analysis of all the metabolites and complementary step-wise dimension reduction would reveal the biochemical pathways involved with the effect you observed.
Response: That is actually a really interesting question and we also considered PC analysis with all metabolites ourselves. Unfortunately, the number of animals included in the study was insufficient to do the analysis with enough statistical power. We used this approach to validate the potential of the 3 metabolites to discriminate between the groups.
3) Do you have any other behavioral data apart from what is presented in Fig 1. (velocity, distance, etc.)? Behavioral protocols differ greatly in the literature so I suggest you describe your behavioral protocol a little bit more in detail. Were the mice removed once they found the platform? What was the cumulative time spent in the pool for all the groups? What do you think why there was increased latency on the first day of the protocol and how are the latency timed distributed in the first trial?
Response:
All mice had an extensive battery of tests including open field behavior in TruScan activity monitoring cages to assess distance, velocity, vertical plane entries and stereotypic moves. Mice were also assessed in the Rotarod test for motor function and grip strength test. For all parameters in these behavior tests there were no differences between wildtype and rTg4510 mice.
In the Morris Water maze test in cued phase, mice that found the platform were allowed to remain on it for 15 seconds before being removed. This was repeated 3 times with a 2 minute interval between trials. We have added this additional information on line 142. Trials were repeated 3 times with a cumulative time in the pool of 90 seconds. In addition, we found no difference in average speed between rTg4510 and non-Tg littermates, indicating no difference between these groups in the ability to swim. This information has been added on line 144 and line 221
The reviewer is correct in pointing out that the rTg4510 mice have a significant increase in latency time on the first trial day. An increase in anxiety, which has been reported in the rTg4510 mouse model (Cook et al. Neurobiology Aging. 2014 Jul;35(7):1769-77), may contribute to a delay in finding the visible platform.
Reviewer 2 Report
The investigators have shown that metabolomic analysis revealed significant differences between WT and rTg4510 AD mouse model with tau pathology. The results give insight to metabolic derangements of AD. The manuscript is in general well written. And the manuscript revealed important findings of differences in metabolomics analysis between WT and Tg AD mice with tauopathy. The experiments are appropriate with reliable findings and the Introduction and Disucssion are well written.
Author Response
The investigators have shown that metabolomic analysis revealed significant differences between WT and rTg4510 AD mouse model with tau pathology. The results give insight to metabolic derangements of AD. The manuscript is in general well written. And the manuscript revealed important findings of differences in metabolomics analysis between WT and Tg AD mice with tauopathy. The experiments are appropriate with reliable findings and the Introduction and Disucssion are well written.
Response: The authors very much appreciate the reviewer’s comments.